# Direct and Label-Free Monitoring of Albumin in 2D Fatty Liver Disease Model Using Plasmonic Nanogratings

**DOI:** 10.3390/nano10122520

**Published:** 2020-12-15

**Authors:** Gerardo A. Lopez-Muñoz, Maria Alejandra Ortega, Ainhoa Ferret-Miñana, Francesco De Chiara, Javier Ramón-Azcón

**Affiliations:** 1Institute for Bioengineering of Catalonia (IBEC), The Barcelona Institute of Science and Technology, Baldiri I Reixac, 10-12, 08028 Barcelona, Spain; glopez@ibecbarcelona.eu (G.A.L.-M.); mortega@ibecbarcelona.eu (M.A.O.); aferret@ibecbarcelona.eu (A.F.-M.); fdechiara@ibecbarcelona.eu (F.D.C.); 2ICREA-Institució Catalana de Recerca i Estudis Avançats, 08010 Barcelona, Spain

**Keywords:** 2D fatty liver in vitro model, Blu-Ray disc, plasmonic nanomaterials, Label-Free Biosensing

## Abstract

Non-alcoholic fatty liver (NAFLD) is a metabolic disorder related to a chronic lipid accumulation within the hepatocytes. This disease is the most common liver disorder worldwide, and it is estimated that it is present in up to 25% of the world’s population. However, the real prevalence of this disease and the associated disorders is unknown mainly because reliable and applicable diagnostic tools are lacking. It is known that the level of albumin, a pleiotropic protein synthesized by hepatocytes, is correlated with the correct function of the liver. The development of a complementary tool that allows direct, sensitive, and label-free monitoring of albumin secretion in hepatocyte cell culture can provide insight into NAFLD’s mechanism and drug action. With this aim, we have developed a simple integrated plasmonic biosensor based on gold nanogratings from periodic nanostructures present in commercial Blu-ray optical discs. This sensor allows the direct and label-free monitoring of albumin in a 2D fatty liver disease model under flow conditions using a highly-specific polyclonal antibody. This technology avoids both the amplification and blocking steps showing a limit of detection within pM range (≈0.26 ng/mL). Thanks to this technology, we identified the optimal fetal bovine serum (FBS) concentration to maximize the cells’ lipid accumulation. Moreover, we discovered that the hepatocytes increased the amount of albumin secreted on the third day from the lipids challenge. These data demonstrate the ability of hepatocytes to respond to the lipid stimulation releasing more albumin. Further investigation is needed to unveil the biological significance of that cell behavior.

## 1. Introduction

The chronic accumulation of fat within the liver, known as non-alcoholic fatty liver (NAFL), increased exponentially in the last 10 years. Over 25% of the global population is affected by NAFL, although it is underdiagnosed [1]. This tremendous upsurge in NAFL disease (NAFLD) prevalence is related to increased consumption of ultra-processed low-cost food and a dramatic decline in physical activity over the last years. NAFL range from simple steatosis, where >5% of the liver cells contain lipid droplets, to its more aggressive form, the non-alcoholic steatohepatitis (NASH), characterized by liver inflammation. The gold standard for detecting NAFLD is the biopsy, where a trained pathologist scores the disease stage [2].

The steatotic hepatocytes (the primary cell type of the liver) have altered cell metabolic capacity and plasticity, leading to a more vulnerable liver to further injuries such as infection and oxidative stress [3]. Moreover, there is no way to detect the early stage of hepatocytes’ response to chronic lipid accumulation. Therefore, the molecular mechanism(s) that underlie the steatosis remains unveiled. The study of the hepatocytes’ metabolism and how they change from healthy to disease status is of primary importance for designing a more precise therapeutic intervention.

Albumin (ALB) is a globular transport protein synthesized in the liver by hepatocytes and released into the bloodstream. ALB participates in many functions within the body, such as regulating the oncotic pressure and boosting the immune system. In addition, it functions as a carrier of a variety of compounds such as hormones, steroids, drugs, and fatty acids. The level of albumin in healthy humans varies from 3.5 g/dL to 5 g/dL, and it represents up to 50% of the total plasma protein content [4]. Finally, the monitoring of ALB levels has enormous clinical relevance because its level is correlated with the liver’s functionality. The serum ALB levels in malnourished people and patients with liver damage are dramatically reduced with the progression of the disease.

Cell culture offers a low-cost and straightforward functional test to study the cells’ response to the lipid challenge. It is a widely used in vitro tool that can contribute to our understanding of cell biology, mechanisms of diseases, drug action, and protein/biomarkers production [5]. Cell culture integration with lab-on-a-chip devices would offer a significant advantage in monitoring the real-time release of potential biomarkers. Optical biosensors are ideal candidates for those kinds of applications because of their label-free detection of multiple analytes with high sensitivity while avoiding unnecessary sample pretreatments [6].

Recently, a simple and cost-effective fabrication method to develop optical biosensors based on plasmonic nanogratings has been demonstrated [7]. This approach is based on industrially produced Blu-ray discs as a nanograting-containing polycarbonate substrate. This process allows the possibility to generate plasmonic effects in the visible range when incorporating gold layer films with thicknesses in the range of 50–100 nm with a high degree of reproducibility and low-cost and straightforward fabrication while achieving biosensing performance similar to those obtained with similar engineered nanostructures [8,9].

Based on these considerations, we have developed a nanostructured plasmonic sensor based on commercial Blu-ray discs with integrated microfluidics for sensitive detection of albumin in cell culture supernatant from a 2D fatty liver disease model (See Figure 1). The left side of Figure 1 shows a biological overview developed in this work. 2D in vitro model of fatty hepatocytes and control were evaluated under different FBS content for up to three days. Collected samples were later interrogated by a home-made setup containing the functionalized gold nanograting-based plasmonic sensors, were fabricated following a simple and reproducible fabrication process, where a direct immunoassay detection was implemented for albumin detection (right side Figure 1). This approach allows direct and label-free monitoring of proteins of interest while avoiding the use of incubation or washing steps and any other sample pretreatment.

## 2. Materials and Methods

### 2.1. Development of the In Vitro Fatty Liver Disease Model

The cells employed for this study are healthy immortalized mouse hepatocytes purchased from ATCC^®^ AML12 (CRL-2254™, Manassas, VI, USA). They were kept in DMEM:F12 Medium (ATCC 30-2006) supplemented with 10% fetal bovine serum (FBS; ATCC 30-2020), 10 µg/mL insulin, 5.5 µg/mL transferrin, 5 ng/mL selenium (41400045, Life Technologies S.A., Frederick, MD, USA), 40 ng/mL dexamethasone (D4902, Merk, FS, Switzerland). For each experiment, the cells were seeded at a density of 26,000/cm^2^ and cultured at 37 °C, 5% CO_2_.

Non-esterified fatty acids (NEFAs) solution is prepared to dissolve oleic acid (OA, O1008-1G, Merk, FS, Schaffhausen, Switzerland) and Palmitic acid (PA, P0500-10G, Merk, FS, Schaffhausen, Switzerland) in PBS with bovine serum (1%) at ratio OA/PA 1:2. The final concentration tested is 400 µM.

Cell metabolism is measured using the CellTiter 96^®^ AQueous One Solution Cell Proliferation Assay (G3582, Promega, AM, Dübendorf, Switzerland). Briefly, at day 1 and day 3, the supernatant was removed, and cells were washed twice with PBS. After that, 100 µL of fresh medium plus 20 µL of MTS was added and incubated for 3 h. The absorbance was recorded at 490 nm.

Cell viability was assessed by alamarBlue™ Cell Viability Reagent (DAL1025, Thermo Fisher Scientific, Spain). At indicated time points, the supernatant was removed, and cells were washed twice with PBS. After that, 90 µL of fresh medium plus 10 µL of alamarBlue™ reagent for 3 h. The fluorescence signal was recorded using Ex(530 nm)/Em(590 nm).

The intracellular lipid uptake was observed using adipored™ (PT-7009, Lonza, Rockville, MD, USA) using fluorescence microscopy Ex(485 nm)/Em(535 nm) while the signal was recorded using a spectrophotometer with the same Ex/Em wavelength. The intracellular level of albumin was assessed by immunocytochemistry. The cells were seeded onto µ-Slide 8 Well (80826, Ibidi, Gräfelfing, Germany) for the indicated length of the experiments. On the day of the assay, the cells were washed twice with PBS. Afterward, they were fixed using 4% paraformaldehyde-PBS at room temperature for 10 min. The permeabilization was obtained using a 0.5% Triton X-100 in PBS solution at room temperature (RT) for 5 min. The blocking was done using 2% horse serum in PBS for 1 h at RT. The albumin antibody (GTX102419, GeneTex, Irvine, CA, USA) was diluted 1:200 in 1% horse serum solution and incubated overnight at 4 degrees. The day after, the excess of the antibody was washed away with PBS, the cells were then incubated with a goat anti-mouse (A28175, Thermo Fisher, Waltham, MA, USA) diluted 1:2000 in PBS in 1% horse serum solution. DAPI was also added at 1:1000 concentration for 1 h at RT. The cells were rewashed with PBS and mounted with Fluoromount-G™ (00-4958-02, Thermo Fisher, Waltham, MA, USA).

### 2.2. Fabrication and Integration of the Nanoplasmonic Chip

Single-layer recordable Blu-ray discs (43743, Verbatim, Taipei, Taiwan) were used after the removal of their protective (polycarbonate) and reflective (aluminum) films. These were removed by cutting the disc in individual plasmonic chips (size 8.5 cm^2^) and then immersing it in a hydrochloric acid solution (2M HCl) overnight. The bared polycarbonate substrates were rinsed with deionized water and nitrogen dried. The chips were covered with a laser cut Mylar stencil sheet (125 microns) as an evaporation mask placed in a vacuum deposition system (Univex 450B, Oerlikon Leybold, Munich, Germany). An 80 nm thick layer of gold was deposited by resistive thermal evaporation (1 Å/s, thickness control via quartz crystal sensor) [7]. A microfluidic flow cell for carrying out the biodetection in solution was developed using a patterned microfluidic channel in a 140 μm thick double-sided adhesive tape sheet (Mcs-foil-008, Microfluidic ChipShop GmbH, Jena, Germany). The proposed design integrates a microfluidic splitter on the chip that allows duplicated biodetection of one protein target or multiplexed biodetection of two protein targets in parallel. A 2 mm tick patterned polymethyl methacrylate (PMMA) lid was added as a cover to facilitate the fluidic tubes’ connection.

### 2.3. Experimental Optical Setup and FDTD Simulations

The integrated sensor chips were clamped to a custom-made optical platform for reflectance measurements. The chips were connected to a microfluidic pressure pump (OB1 Mark I, Elveflow, Paris, France), with adjustable pumping speed guaranteeing a constant liquid flow. Reflectance measurements were performed under TM-polarization of a compact stabilized broadband light source (SLS201L, Thorlabs, Bergkirchen, Germany) at 60°. The incident excitation plane was perpendicularly aligned to the nanograting direction. The reflected light was collected and fiber-coupled to a compact Charge-Coupled Device (CCD) spectrometer (Exemplar UV-NIR, BWTek, Lübeck, Germany).

Reflectivity spectra were acquired every 1 ms, and 50 consecutive spectra were measured and averaged to provide the final spectrum. These acquisition parameters were selected to obtain the optimum signal to noise (S/N) ratio without significantly increasing the data acquisition time. Changes in the resonance peak position (λSPR) were tracked via centroid determination by peak analysis using Origin 2018 software (OriginPro, OriginLab Co., Northampton, MA, USA). Reflection spectra were collected in water (n = 1.33 RIU) and solutions of glucose in water (ranging between 1.337 and 1.368 RIU) to determine the bulk sensitivity of the plasmonic nanograting sensor.

Three-dimensional FDTD simulations were performed using commercial software (FDTD solution, Lumerical, Inc., Vancouver, BC, Canada). Structural parameters of the Blu-ray discs based on Atomic Force used in the simulations (i.e., a grating period of 320 nm, grating width of 100 nm, and a height of 20 nm) were according to Atomic Force Microscopy (AFM) images from previous results [10]. Periodic boundary conditions were used on the x and *y*-axis, and the perfectly matched layers (PML) approach was used on the *z*-axis, with a uniform mesh size of 2 nm in all axis. The FDTD simulations were performed in the range from 400 nm to 1000 nm, under TM-polarized light with an oblique light incidence angle of 60°.

### 2.4. Surface Biofunctionalization of Nanoplasmonic Sensing Platform

Alkanethiol for self-assembled monolayer (SAM) formation (11-mercaptoundecanoic acid, MUA), (1- ethyl-4 (3-dimethylaminopropyl) carbodiimide hydrochloride (EDC) and sulfo-N hydroxysuccinimide (s-NHS) for carboxylic groups activation and ethanolamine were acquired to Sigma Aldrich (Darmstadt, Germany). Sensor chips were cleaned and activated for surface functionalization by performing consecutive rinsing with ethanol and deionized water, drying with N_2_ stream, and finally by placing them in a UV ozone generator (BioForce Nanoscience, Golden Aspen, IA, USA) for 20 min. An alkanethiol self-assembled monolayer (SAM) with reactive carboxylic groups was obtained by coating the sensor chip with 2.5 mM MUA in ethanol overnight at room temperature. Then, the surface was rinsed with ethanol and carefully washed with MES buffer.

The immobilization of the Mouse Serum Albumin antibody (MSA) was performed ex-situ following the protocol previously described by Acimovic-Ortega et al. [11]. For the activation of the carboxylic groups, a solution of 0.2 M EDC/0.05 M sulfo-NHS in (2-ethanesulfonic acid) MES buffer (25 mM pH 5.7) was dropped over the SAM monolayer for 40 min. After washing steps using MES buffer, 50 µg/mL of MSA antibody solution in (phosphate buffer) PB (10 mM pH 7.4) was dropped for 120 min. Finally, the sensor was submerged on an ethanolamine solution 50 µg/mL prepared in PB 10 mM pH 7.2 for 15 min to block unreacted remaining active carboxylic groups. Chips were carefully dried with an N_2_ stream and bonded to the microfluidic channels, and finally placed in the optical platform and filled with PB for optimization and assessment studies. Different albumin protein concentrations diluted in Phosphate Buffer Saline 10 mM pH 7.2 (PBS) flowed over the functionalized surface at 80 µL/min. Calibration curves were fitted to a one-site specific binding model. Data points for each concentration were collected 30 min after injection. Albumin from mouse serum (A-3139, Merck, Darmstadt, Germany) was used as a standard protein for the preparation of calibration curves at different FBS content and sensor optimization.

## 3. Results and Discussion

### 3.1. In Vitro Fatty Liver Disease Model

To reproduce some of the features of NAFLD in vitro, we first prepared the lipid solution (Non-esterified fatty acids—NEFAs) dissolving oleic and palmitic acids in a solution of bovine serum and phosphate buffer saline (PBS) because of their insolubility in water solutions as previously published [12]. Second, we challenged the hepatocytes with a lipid solution concentrated 400 µM at 0, 1, and 3 days using 0%, 2%, and 10% of FBS in order to see which of those conditions accumulate the highest concentration of lipid without affecting the cell viability (Figure 2a) significantly. From day 1, all the experimental conditions displayed lipid droplets accumulation using bright microscopy (yellow arrows, Figure 2b). The NEFAs are internalized by the cells and processed as droplets of triglycerides (a zoomed image is shown in A1). We employed two tests to assess cell viability and metabolism: the MTS (absorbance) and Alamarblue™ (Thermo Fisher, Waltham, MA, USA) (fluorescence), respectively. Both assays showed an increase in cell number and metabolism from day 1 (D1) to day 3 (D3) in the untreated conditions cultured in the presence of a different concentration of FBS (Figure 2c). However, in 0% FBS, cells cultured showed a smaller increment over time than FBS 2%, while FBS 10% showed the highest increment. Those tests also revealed a significant reduction in cells’ viability and metabolism upon lipid exposure in all the conditions under investigation (Figure 2c). The lipid accumulation within the cells was assessed using AdipoRed™ assay (Figure 2d). The AdipoRed™ becomes fluorescent when stored in a hydrophobic environment like a cell membrane containing double-layer lipids. In the fat-treated cells, the AdipoRed™ diffuses through the cell membrane, and it accumulates within the cytoplasmatic lipid droplets (yellow arrows Figure 2e). All the conditions under investigation exhibit a significant accumulation of AdipoRed™ upon treatment with NEFAs solution compared with the untreated condition, regardless of the cell number and metabolism (yellow arrows, Figure 2e). Interestingly, starved cells (FBS 0%) did not show almost any fat accumulation upon treatment. These data show that the fat exposition induces a reduction in hepatocytes viability and metabolism and a concomitant intracellular lipid accumulation. Both are features of the NAFLD liver.

### 3.2. Simulation and Characterization of Nanoplasmonic Sensor

Gold-coated nanogratings were fabricated, taking advantage of the precise and large area of nanostructured arrays present in commercial Blu-ray discs (see Figure 3a for scanning electron image). As shown in Figure 3b, similar behavior is calculated between the FDTD and the experimental reflectance spectra. The differences in the resonance peaks position between the simulated and fabricated nanograting are mainly due to geometrical differences in the simulations (nanogratings are radial to the Blu-ray disc’s circumference, differences in dimensions and shape) and metallic layer imperfections. The FDTD electric field distribution for an 80 nm gold-coated nanograting (Figure 3c) under a 60° angle of light incidence shows a minimal interaction of plasmons with the underlying substrate. It results in a high intensity and long decay length of the electric field over the sensing media due to the generation of surface lattice resonances, as previously reported [7]. A bulk sensitivity of ≈392 nm/RIU (Figure 3d,e) is achieved, which is highly competitive and similar to those obtained with engineered nanoslits/nanogratings sensors, making them an interesting development of cost-effective biosensor devices.

### 3.3. Albumin Detection by the Nanoplasmonic Sensing Integrated Platform

Once nanoplasmonic sensors were fabricated, the MSA was anchored to the gold surface to perform a selective detection of albumin via self-assembled monolayer formation. After the ex-situ biofunctionalization, a connection with the PMMA microfluidics cover was performed. Figure 4a shows the nanostructured plasmonic biosensor integrated with the microfluidic splitter on the chip. This design simplifies the microfluidic instrumentation and, if required, a multiplexed measurement (either the biodetection of two biomarkers or the replica of a measurement of a single biomarker). The insert in Figure 4a shows a scanning electron microscopy (SEM) image of the fabricated gold nanogratings with high reproducibility on a large scale due to industrial manufacturing. Figure 4b shows the nanoplasmonic sensor connected to the microfluidic system in reflectance mode experimental setup and clamped to a motorized linear stage to move between the sensing areas.

Optimization of the capture antibody concentration is shown in Figure 4c. The black line corresponds to the SPR shift for MSA antibody at different concentrations after the ex-situ biofunctionalization protocol. 10, 20, 50, and 100 µg/mL was evaluated, obtaining a saturation plateau for concentration up to 20 µg/mL. Nevertheless, to evaluate the detection’s capability for the anchored antibody layers, it flowed 1µg/mL albumin protein solution. In Figure 4c (red line graph), we can observe that maximum SPR shift signal is obtained for 50 µg/mL MSA concentration, which we assume as optimal concentration for further experiments.

Calibration curves were performed using serial dilutions of commercial albumin 0.1, 1, 10, 200, 1000, and 5000 ng/mL diluted in cell media containing FBS at 0, 2, and 10%. Before flowing protein solutions, cell media without albumin was used to set a baseline, subtracting the possible matrix effects. In Appendix A(A2) in Appendix A, it can be observed the native SPR peak and shift generated by the increment of albumin concentration using an additive assay.

Curves were fitted to a four-parameter equation according to the following formula: Y = [(A − B)/1 − (x/C)D] + B, where A is the maximal signal, B is the minimum, C is the concentration producing 50% of the maximal signal, and D is the slope at the inflection point of the sigmoid curve. The calculation of the limit of detection (LOD) and the limit of quantification (LOQ) was based on ELISA methodology [13]. For a standard curve, the LOD is considered the concentration corresponding to the interpolated intersection of the lower asymptote’s 10% confidence interval with the four-parameter equation fit of standards data. In Figure 4d, it is shown the calibration curve performed in cell media containing 2% FBS. After analysis, we obtained a LOD of 0.26 ng/mL (≈5 pM). The obtained LOD is at least one order of magnitude lower to those previously obtained by conventional plasmonic biosensors [14,15], one of them in the detection of albumin in cell cultures [15], and other optical [16,17], electrochemical [18,19], and magnetic-based [20] biosensors. Calibration curves using 0 and 10% FBS are shown in Appendix A, obtaining LOD of 0.22 and 0.23 ng/mL, respectively. FBS content does not have a direct effect on albumin detection. Different experiments were performed with a total of two replicates. Those calibration curves were used later during quantification assays for unknown samples generated from the in vitro system. As previously reported by López-Muñoz et al. [7], it is possible to achieve nanoplasmonic sensors with a coefficient of variation below 1% in their optical performance with this fabrication process, demonstrating the high reproducibility between them.

Detection of secreted albumin from the in vitro model of fatty liver was performed under fluidic conditions. Supernatants of controls and fatty liver was collected on day 0, 1, and 3. Each sample was interrogated by individual 2-channel nanoplasmonic sensing platforms and later fitted from the corresponding calibration curve for quantification. Figure 5a shows the albumin secretion profile from untreated and fatty hepatocytes for all the conditions. The albumin concentration increases more significantly in the 2% FBS condition compared to the control.

Nevertheless, the albumin secretion remains constant at FBS content of 0 and 10%. In the case of 2% FBS condition, we independently evaluate albumin secretion of samples in cell media with and without fat to confirm that the signal’s increment is selective in detecting the protein without added effect from the fat supplement media supplements (Figure 5b). We validated the data obtained from the sensors with intracellular albumin level by immunocytochemistry (Figure 5c). The cells grew more in the presence of FBS 10% than the cells with the other two FBS content; however, they secreted less albumin than the other two, especially the FBS 2%. The cells seem to find an optimal starving environment with FBS 2% for albumin production because they differentiate more than the other two conditions.

## 4. Conclusions

The in vitro data presented reflect both the phenotypical and functional changes in fatty hepatocytes in vivo. The viability and metabolic activity decrease upon challenge with NEFAs while the albumin production increases due to adaptation to the new environment. The simple label-free integrated plasmonic biosensor based on Blu-ray discs nanogratings allowed us to monitor albumin secretion overtime in an in vitro fatty liver model. The plasmonic biosensor represents a useful tool to study the evolution of the disease in vitro. Moreover, we took advantage of the cost-effective Blu-ray disc to achieve the fabrication of high throughput sensors. This prototype is customizable, and it has been employed as a multi-analyte detection system in cell culture supernatant media, with a LOD in the pM order without any amplification or pretreatment of the sample. Due to its high versatility and the straightforward integration in lab-on-a-chip devices, the presented plasmonic nanomaterial is a promising candidate for developing monitoring platforms for cell cultures.

## Figures and Tables

**Figure 1 nanomaterials-10-02520-f001:**
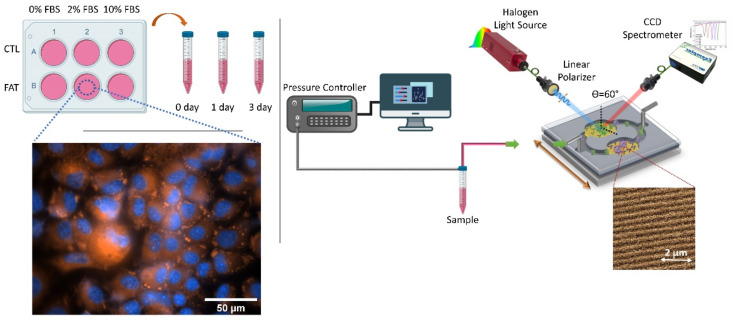
Schematic representation for developing a high throughput plasmonic sensing platform for monitoring albumin levels in 2D in vitro model of fatty liver disease.

**Figure 2 nanomaterials-10-02520-f002:**
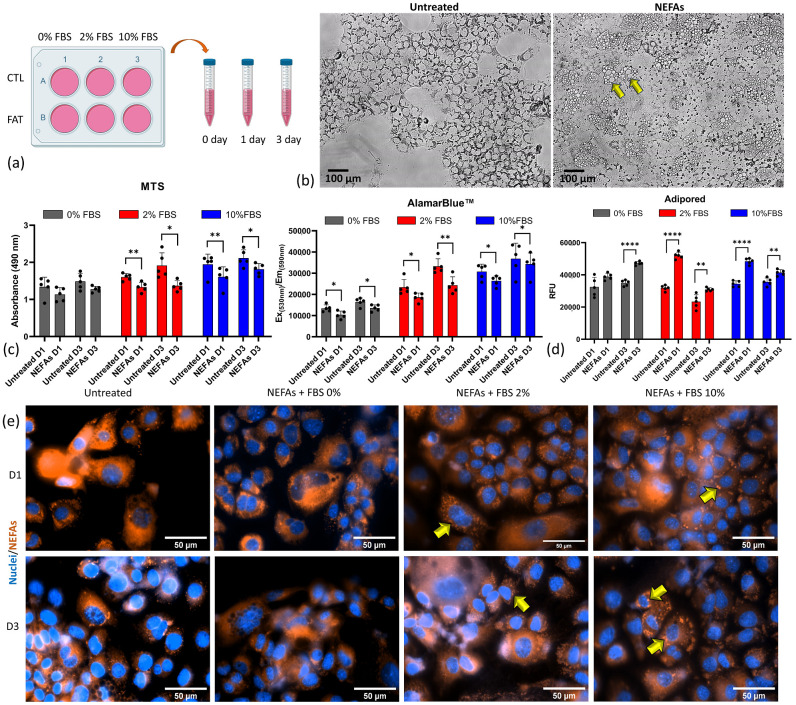
In vitro model of fatty hepatocytes. (**a**) experimental setup. (**b**) Bright microscopy of mouse hepatocytes with and without treatment with non-esterified fatty acids. (**c**) Cell viability and metabolism of hepatocytes treated with 400 µM of NEFAs up to three days; * *p* ≤ 0.05; ** *p* ≤ 0.01; *** *p* ≤ 0.001; **** *p* ≤ 0.0001. (**d**) Quantitative and (**e**) qualitative intracellular fat accumulation using Adipored™ assay. Yellow arrows indicate the lipid droplets accumulated inside the cells, especially around the nucleus.

**Figure 3 nanomaterials-10-02520-f003:**
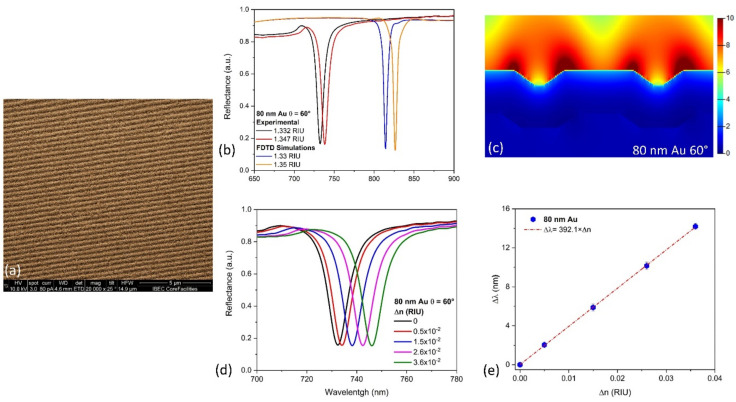
FDTD simulations of the proposed sensor and sensing performance evaluation. (**a**) SEM photograph of the fabricated plasmonic nanogratings. (**b**) Experimental and simulated optical reflectance spectra under TM-polarization with a light incident angle of 60° for the nanostructured plasmonic sensor fabricated. (**c**) Simulated electric field distribution for the 80 nm gold thickness layer on the polycarbonate nanograting under TM-polarization for a light incidence angle of 60°. (**d**) Displacement of the reflectance spectra for the plasmonic nanogratings based sensor fabricated with different refractive index solutions based on ethanol-water mixtures. (**e**) Calibration curve and bulk sensitivity determination for the 80 nm gold thickness layer sensor at a light incidence angle of 60°.

**Figure 4 nanomaterials-10-02520-f004:**
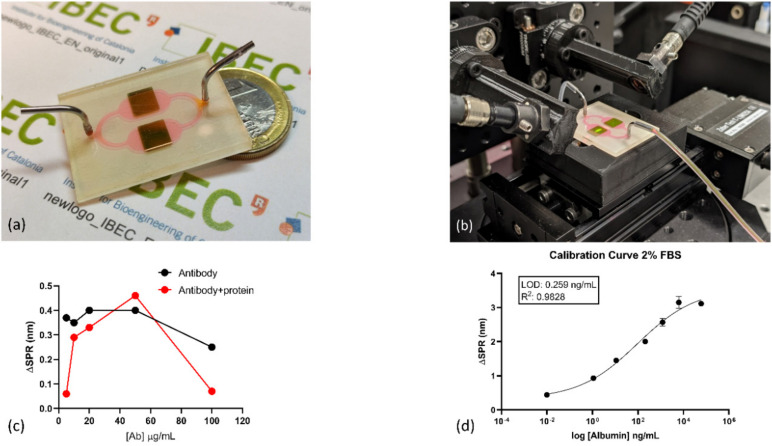
Experimental setup and optimization of biosensing assay. (**a**) Photograph of the 2-channel packed sensor. (**b**) Photograph of the experimental system. (**c**) Mouse serum albumin (capture antibody) optimization. The black line corresponds to SPR shift obtained by changing the MSA concentration on the functionalized nanoplasmonic surface; Redline corresponds to SPR shift obtained after evaluating detection of 1µg/mL albumin at the previous MSA concentrations anchored on sensors surface (**d**) calibration curve of albumin detection performed in DMEM cell media containing 2%FBS. LOD of 0.26 ng/mL was obtained.

**Figure 5 nanomaterials-10-02520-f005:**
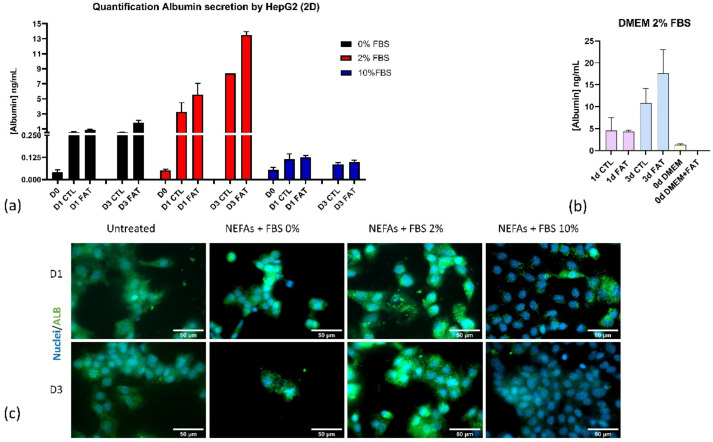
Comparison biodetection of albumin by nanoplasmonic biosensor and fluorescence assay. (**a**) Quantification of secreted albumin generated by 2D in vitro model of fatty hepatocytes treated with cell media at 0, 2, and 10% FBS during three days. (**b**) Evaluation of matrix effect and fat supplement in the cell media with the detection of secreted albumin performed in cell media at 2% FBS. (**c**) Intracellular staining of albumin using immunochemistry technique.

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
