# Peer review of "Direct and Label-Free Monitoring of Albumin in 2D Fatty Liver Disease Model Using Plasmonic Nanogratings"

_nanomaterials, 2020, doi:10.3390/nano10122520_

Round 1
Reviewer 1 Report
Review of the article “Direct and Label-Free Monitoring of Albumin in 2D Fatty Liver Disease Model using Plasmonic Nanogratings” by G. A. Lopez-Muñoz, Ma. A. Ortega, A. Ferret-Miñana, F. De Chiara and J. Ramón-Azcón, Nanomaterials, Manuscript ID nanomaterials-1024759.
In this article, the fabrication of an integrated plasmonic biosensor based on gold nanogratings is described. This lab-on-a-chip system uses commercial Blu-ray discs as substrates for the deposition of a gold thin layer film by resistive thermal evaporation. This gold layer is subsequently derivatised with mouse serum albumin antibody, which acts as bioreceptor for the detection of albumin levels in a 2D fatty liver disease model. The authors optimized the experimental conditions for the fatty liver disease model, assessed cell viability and metabolism via MTS and Alamarblue™ assay, respectively, and confirmed the validity of their model. After calibration with standard solutions of albumin, the biosensor exhibited a limit of detection of 0.39 ng/mL. The results obtained with the use of this biosensor were found to be in line with the data obtained via Alamarblue™ assay.
Major comments:
The idea of the fabrication of this label-free biosensor is very interesting, both for its cost-effectiveness and for its miniaturized nature. Nevertheless, some issues about applicability have to be addressed by the authors:
- The sensitivity of the biosensor does not seem to be high: according to Fig. 4b, an increase in albumin concentration from 1 ng/mL to circa 100 ng/mL induces a ΔSPR = 1 nm. On the other hand, in Figure 5a is possible to detect variations in the order of 1 ng/mL. How do the authors explain this contrasting behaviour?
- I strongly suggest to add an SEM image of the biosensor, showing its structure.
- Lines 209-212: did the authors check if the significant difference (circa 80 nm) between simulated and experimental reflectance peaks (Fig. 3a) could origin from the hybridization of plasmonic modes due to the near-field coupling occurring between gold nanoparticles? See as reference Franklin, D. et al. Self-assembled plasmonics for angle-independent structural color displays with actively addressed black states. Natl. Acad. Sci. U. S. A. 117, 13350-13358, doi:10.1073/pnas.2001435117 (2020).
- Lines 254-256: how did the author derive the fitting equation?
- Figure 2d: which method did the authors use for the quantification in Adipored™ assay? According to the representative fluorescence images, it seems that the orange fluorescence decreases increasing the concentration of FBS in day 1, which is the opposite of the results shown in the column graph on the left.
- The authors should definitely expand the bibliography and compare the performance of the proposed biosensors with the others used for albumin detection in the relevant literature.
Minor comments:
- Line 63: please add a comma after “recently”
- Line 70: please add a comma after “considerations”
- Line 106: please use “Afterwards” instead of “Following”
- Line 118: 2 should be superscript in “cm2”
- Line 155 + 167: 2 should be subscript in “N2”
- Line 168-169: please substitute “MSA protein” with “albumin”
- Line 190: please write “AdipoRed™”
- Line 197: please substitute “while” with “and a concomitant”
- Line 271: space missing between “(a)” and “Photograph”
- Line 274: please substitute “red” with “black”
- Line 276: please add “albumin” after 1 μg/mL
- Line 295: please delete g in “albuming”
- Line 299: please change to “immunochemistry”
Author Response
Review of the article “Direct and Label-Free Monitoring of Albumin in 2D Fatty Liver Disease Model using Plasmonic Nanogratings” by G. A. Lopez-Muñoz, Ma. A. Ortega, A. Ferret-Miñana, F. De Chiara and J. Ramón-Azcón, Nanomaterials, Manuscript ID nanomaterials-1024759.
In this article, the fabrication of an integrated plasmonic biosensor based on gold nanogratings is described. This lab-on-a-chip system uses commercial Blu-ray discs as substrates for the deposition of a gold thin layer film by resistive thermal evaporation. This gold layer is subsequently derivatised with mouse serum albumin antibody, which acts as bioreceptor for the detection of albumin levels in a 2D fatty liver disease model. The authors optimized the experimental conditions for the fatty liver disease model, assessed cell viability and metabolism via MTS and Alamarblue™ assay, respectively, and confirmed the validity of their model. After calibration with standard solutions of albumin, the biosensor exhibited a limit of detection of 0.39 ng/mL. The results obtained with the use of this biosensor were found to be in line with the data obtained via Alamarblue™ assay.
Major comments:
The idea of the fabrication of this label-free biosensor is very interesting, both for its cost-effectiveness and for its miniaturized nature. Nevertheless, some issues about applicability have to be addressed by the authors:
- The sensitivity of the biosensor does not seem to be high: according to Fig. 4b, an increase in albumin concentration from 1 ng/mL to circa 100 ng/mL induces a ΔSPR = 1 nm. On the other hand, in Figure 5a is possible to detect variations in the order of 1 ng/mL. How do the authors explain this contrasting behaviour?
Answer: Thanks to the reviewer for the observation; we are able to detect low concentrations ≈ 1ng/mL mainly to the following facts: Although conventional plasmonic biosensors have a higher “bulk sensitivity” compared to nanoplasmonic sensors, nanoplasmonic sensors have a higher surface sensitivity, which allows them to detect an analyte at a concentration up to an order of magnitude lower compared to conventional plasmonic sensors. Špačková, B., Lynn Jr, N. S., Slabý, J., Šípová, H., & Homola, J. (2018). A route to a nanoplasmonic biosensor’s superior performance: consideration of both photonic and mass transport aspects. Acs Photonics, 5(3), 1019-1025.
On the other hand, the biosensor’s final LoD is also correlated with the spectrometer’s resolution and the detection algorithm. In our case, it is based on centroid detection. In previous results, we have been able to detect refractive index changes in the order of ≈ 1x10-5 RIU using a similar experimental set-up and the centroid detection algorithm, which corresponds to a ∆λ below 0.01 nm. Ref. [7]
- I strongly suggest to add an SEM image of the biosensor, showing its structure.
Answer: Thanks for the suggestion; we have included an SEM image in Figure 3a.
- Lines 209-212: did the authors check if the significant difference (circa 80 nm) between simulated and experimental reflectance peaks (Fig. 3a) could origin from the hybridization of plasmonic modes due to the near-field coupling occurring between gold nanoparticles? See as reference Franklin, D.et al.Self-assembled plasmonics for angle-independent structural color displays with actively addressed black states. Natl. Acad. Sci. U. S. A. 117, 13350-13358, doi:10.1073/pnas.2001435117 (2020).
Answer: We are glad that the reviewer made this question. The significant difference between the experimental and simulation reflectance peaks is mainly due to differences in the nanograting’s dimensions and shape in the simulations. They are also radial to the circumference of the Blu-Ray disc, while in the simulations, we considered straight nanogratings to simplify the simulation process. Although it is not perceptible at the nanoscale, it likely influences the plasmonic band position on a larger scale. An AFM image where this can be observed is present in the following article: Hagemeier, S., Schake, M., & Lehmann, P. (2019). Sensor characterization by comparative measurements using a multi-sensor measuring system. Journal of Sensors and Sensor Systems, 8(1), 111-121. This described phenomenon is mainly present in metal-insulator-metal configurations; it is a couple of localized surface plasmon resonances and cavity mode. We are expected to take advantage of this phenomenon in future works.
We have rewritten this part in the paragraph (lines 211-215).
- Lines 254-256: how did the author derive the fitting equation?´
Answer: Dear reviewer, as it was described in lines 259-265, the standard curves were fitted to a four-parameter equation according to the following formula:
Y = [(A - B)/1 - (x/C)D] + B
A is the maximal signal, B is the minimum, C is the concentration producing 50% of the maximal signal, and D is the slope at the sigmoid curve’s inflection point. The calculation of the limit of detection (LOD) and the limit of quantification (LOQ) was based on ELISA methodology (Techniques and Instrumentation in Analytical Chemistry 21:287-339, DOI: 10.1016/S0167-9244(00)80013-X, Immunochemical Determination of Pharmaceuticals and Personal Care Products as Emerging Pollutants, DOI: 10.1007/b98616). For a standard curve, the LOD is the concentration corresponding to the interpolated intersection of the 10% confidence interval of the lower asymptote with the four-parameter equation fit of standards data. The software used for this fitting was GraphPad Analysis Software using a dose-response four-parameter fitting model.
- Figure 2d: which method did the authors use for the quantification in Adipored™ assay? According to the representative fluorescence images, it seems that the orange fluorescence decreases increasing the concentration of FBS in day 1, which is the opposite of the results shown in the column graph on the left.
Answer: Thanks to the reviewer for the comment. The Adipored™ signal was recorded by fluorescence (Ex(485 nm)/Em(535 nm) as specified in the Materials and Methods section (line 103-104). The peculiarity of the Adipored™ is to become fluorescent when in contact with a hydrophobic environment like the lipid droplets and the double membrane layer. For this reason, we compared the overall signal between the conditions under investigation by fluorimeter and not by eyes.
- The authors should definitely expand the bibliography and compare the performance of the proposed biosensors with the others used for albumin detection in the relevant literature.
Answer: we want to thank the reviewer; we have compared the performance with different proposed biosensors for albumin detection. We have rewrite part of the text (lines 266-269).
“After analysis, we obtained a LOD of 0.39 ng/mL (≈ 6 pM), the resulted LOD is at least one order of magnitude lower to those previously obtained by conventional plasmonic biosensors [14,15] one of them in the detection of albumin in cell cultures [15], and other optical [16,17], electrochemical [18,19] and magnetic-based [20] biosensors. ”
Minor comments:
- Line 63: please add a comma after “recently”
- Line 70: please add a comma after “considerations”
- Line 106: please use “Afterwards” instead of “Following”
- Line 118: 2 should be superscript in “cm2”
- Line 155 + 167: 2 should be subscript in “N2”
- Line 168-169: please substitute “MSA protein” with “albumin”
- Line 190: please write “AdipoRed™”
- Line 197: please substitute “while” with “and a concomitant”
- Line 271: space missing between “(a)” and “Photograph”
- Line 274: please substitute “red” with “black”
- Line 276: please add “albumin” after 1 μg/mL
- Line 295: please delete g in “albuming”
- Line 299: please change to “immunochemistry”
Answer: Dear reviewer we have already fixed in the new version all the typos.
Reviewer 2 Report
In this paper, Lopez-Munoz and coworkers describe the development of a biosensor platform to monitor albumin secretion form hepatocytes. This is a significant goal because of the association of albumin concentrations with non-alcoholic fatty liver disease (NAFLD). Their approach to creating a successful biosensor was the implementation of a plasmonic grating fabricated from a Blue-ray disk template. With this biosensor albumin was detected with a LOD in the pM range. The biosensor also allowed the optimization of FBS concentration to maximize lipid accumulation in cells.
This work is comprehensive and publication is warranted after major revision, provided the authors address the comments and concerns listed below.
1. In the Materials and Methods section “2.2: Fabrication and integration of the nanoplasmonic chip” more details are necessary for the extraction of the template from the Blue-ray disk. After dissolving the other disk layers in 2 M HCl, what are the steps for handling the relevant layer? What material is the relevant layer composed of? Is this layer mounted on a substrate prior to gold deposition? How is the deposited gold thickness measured? If these details were given in a prior publication from this group (Ref [7]), this paper should be cited as well.
2. In Fig. 1b the arrows are said to denote fatty acid accumulation. First, these arrows are tiny. Make them larger. Second, how are the features identified by the arrows indicative of fatty acid accumulation? What should a reader be looking for here? Without being guided a bit, I find it hard to see any difference in these images. Perhaps changing the scale of the images would help too. Zoom in to show more fine detail in the cells.
3. The SEM images in Fig. 1 and Fig. 4 might be better placed as a part of Fig. 3. This would allow the reader to quickly compare the dimensions in the simulation (Fig. 3b).
4. There is a large difference in the resonance wavelength between the measured spectra and the simulated spectra in Fig. 3a. On lines 209-212 the authors state,
“As showed in Figure 3a, there is a good agreement between the FDTD calculated and the experimental reflectance spectra. The differences in the resonance peaks position between the simulated and fabricated nanograting are mainly due to unavoidable geometrical and metallic layer imperfections.”
While broadening of the resonance peak could be attributed to imperfections in the metallic layer, the resonance wavelengths in the experimental and simulated spectra differ by more than 80 nm. This is not “good agreement.” A better explanation for why the experimental and simulated spectra are so different must be offered. Also, what is meant by “geometrical imperfections”? Does this mean that all chips derived from a single Blue-ray disk have variable grating parameters? Or does it mean there is disk-to-disk variation in the grating parameters?
Furthermore, how is it possible to attribute the simulated/experimental spectra difference to variance in metallic coatings and grating geometry, when on lines 266-268 the authors state,
“it is possible to achieve nanoplasmonic sensors with a coefficient of variation below 1% in their optical performance with this fabrication process which demonstrate the high reproducibility between the sensors.”
If the sensor chips are reproducible with small CoV, but their spectra do not agree with simulations, this suggests that the simulations may be inaccurate.
5. What is the source and product number of the commercially-acquired albumin used in these studies, particularly the dilution series described on lines 249-250? This is critical information given the availability of different types of albumin.
6. The calibration curve was fit to an equation shown on lines 254-255. Where does this equation come from? Was it derived by the authors? If it was initially described elsewhere, a citation should be given.
7. The calibration curve in Fig. 4d is plotted as (delta)SPR vs. log [Albumin]. I would caution the authors against plotting signal vs. log concentration. A detailed discussion of why this is not best practice is provided in the recent Perspective article in Analytical Chemistry (P.L. Urban, Analytical Chemistry, 2020, 92, 10210-10212. DOI: 10.1021/acs.analchem.0c02096).
8. There are a few typos that need to be corrected.
-Line 92: Testes?
-Line 105: “essay” should be “assay”
-Line 118: cm2. The “2” should be superscript
-Line 155 and elsewhere: N2. “2” should be subscript
-Line 266: “(2017, Biosens. Bioelectron.)” delete this parenthetical expression and cite Ref. [7].
-Line 191-192. Rewrite this sentence. As written it appears to indicate that the cell membrane contains fat droplets.
Author Response
In this paper, Lopez-Munoz and coworkers describe the development of a biosensor platform to monitor albumin secretion form hepatocytes. This is a significant goal because of the association of albumin concentrations with non-alcoholic fatty liver disease (NAFLD). Their approach to creating a successful biosensor was the implementation of a plasmonic grating fabricated from a Blue-ray disk template. With this biosensor albumin was detected with a LOD in the pM range. The biosensor also allowed the optimization of FBS concentration to maximize lipid accumulation in cells.
This work is comprehensive and publication is warranted after major revision, provided the authors address the comments and concerns listed below.
- In the Materials and Methods section “2.2: Fabrication and integration of the nanoplasmonic chip” more details are necessary for the extraction of the template from the Blue-ray disk. After dissolving the other disk layers in 2 M HCl, what are the steps for handling the relevant layer? What material is the relevant layer composed of? Is this layer mounted on a substrate prior to gold deposition? How is the deposited gold thickness measured? If these details were given in a prior publication from this group (Ref [7]), this paper should be cited as well.
Answer: Dear reviewer, we appreciate suggestions and comments; we have expanded the described information according to comments (lines 117-123): “Single-layer recordable Blu-ray discs (43743, Verbatim, Taiwan) were used after removal of their protective (polycarbonate) and reflective (aluminum) films. These were removed by cutting the disc in individual plasmonic chips (size 8.5 cm2) and then immersing it in a hydrochloric acid solution (2M HCl) overnight. The bared polycarbonate substrates were rinsed with deionized water and nitrogen dried. The chips were covered with a laser cut Mylar stencil sheet (125 microns) as an evaporation mask placed in a vacuum deposition system (Univex 450B, Oerlikon Leybold, Germany). An 80 nm thick layer of gold was deposited by resistive thermal evaporation (1 Å/s, thickness control via quartz crystal sensor) [7].
- In Fig. 1b the arrows are said to denote fatty acid accumulation. First, these arrows are tiny. Make them larger. Second, how are the features identified by the arrows indicative of fatty acid accumulation? What should a reader be looking for here? Without being guided a bit, I find it hard to see any difference in these images. Perhaps changing the scale of the images would help too. Zoom in to show more fine detail in the cells.
Answer: Thanks to the reviewer for the comment. We made the arrows bigger and rearranged the figure 2. We understand the concern of the reviewer about the description of the figure 2e. We added additional information in the text (line 197-198) and in the figure legend (line 210-211). Finally, we decided to leave the images as they are to give the reader the overview of what is happening to the cells.
- The SEM images in Fig. 1 and Fig. 4 might be better placed as a part of Fig. 3. This would allow the reader to quickly compare the dimensions in the simulation (Fig. 3b).
Answer: Dear reviewer we really appreciate your suggestions and comments; we have included a SEM image Figure 3a.
- There is a large difference in the resonance wavelength between the measured spectra and the simulated spectra in Fig. 3a. On lines 209-212 the authors state,
“As showed in Figure 3a, there is a good agreement between the FDTD calculated and the experimental reflectance spectra. The differences in the resonance peaks position between the simulated and fabricated nanograting are mainly due to unavoidable geometrical and metallic layer imperfections.”
While broadening of the resonance peak could be attributed to imperfections in the metallic layer, the resonance wavelengths in the experimental and simulated spectra differ by more than 80 nm. This is not “good agreement.” A better explanation for why the experimental and simulated spectra are so different must be offered. Also, what is meant by “geometrical imperfections”? Does this mean that all chips derived from a single Blue-ray disk have variable grating parameters? Or does it mean there is disk-to-disk variation in the grating parameters?
Furthermore, how is it possible to attribute the simulated/experimental spectra difference to variance in metallic coatings and grating geometry, when on lines 266-268 the authors state,“it is possible to achieve nanoplasmonic sensors with a coefficient of variation below 1% in their optical performance with this fabrication process which demonstrate the high reproducibility between the sensors.”If the sensor chips are reproducible with small CoV, but their spectra do not agree with simulations, this suggests that the simulations may be inaccurate.
Answer: Dear reviewer, we appreciate suggestions and comments; the significant difference between the experimental and simulation reflectance peaks is mainly owing to differences in dimensions and shape of the nanograting in the simulations. The nanogratings are also radial to the circumference of the Blu-Ray disc, while in the simulations, we considered straight nanogratings to simplify the simulation process. Although it is not perceptible at the nanoscale, it likely influences the plasmonic band position on a larger scale. An AFM image where this can be observed is present in the following article: Hagemeier, S., Schake, M., & Lehmann, P. (2019). Sensor characterization by comparative measurements using a multi-sensor measuring system. Journal of Sensors and Sensor Systems, 8(1), 111-121.
We have rewrite part of the paragraph in lines 211-215 “As showed in Figure 3a, there is a similar behavior between the FDTD calculated and the experimental reflectance spectra. The differences in the resonance peaks position between the simulated and fabricated nanograting are mainly due to geometrical differences in the simulations (nanogratings are radial to the circumference of the Blu-ray disc, differences in dimensions and shape) and metallic layer imperfections.”
- What is the source and product number of the commercially-acquired albumin used in these studies, particularly the dilution series described on lines 249-250? This is critical information given the availability of different types of albumin.
Answer: The commercial albumin used as a standard for calibration curves is Albumin from Mouse serum (A3139 – from Sigm Aldrich). This information is described in lines 175, 176, and 177 of the submitted manuscript.
- The calibration curve was fit to an equation shown on lines 254-255. Where does this equation come from? Was it derived by the authors? If it was initially described elsewhere, a citation should be given.
Answer: Dear reviewer, as it was described in lines 259-265, the standard curves were fitted to a four-parameter equation according to the following formula:
Y = [(A - B)/1 - (x/C)D] + B
A is the maximal signal, B is the minimum, C is the concentration producing 50% of the maximal signal, and D is the slope at the sigmoid curve’s inflection point. The calculation of the limit of detection (LOD) and the limit of quantification (LOQ) was based on ELISA methodology (Techniques and Instrumentation in Analytical Chemistry 21:287-339, DOI: 10.1016/S0167-9244(00)80013-X, Immunochemical Determination of Pharmaceuticals and Personal Care Products as Emerging Pollutants, DOI: 10.1007/b98616). For a standard curve, the LOD is the concentration corresponding to the interpolated intersection of the 10% confidence interval of the lower asymptote with the four-parameter equation fit of standards data. The software used for this fitting was GraphPad Analysis Software using a dose-response four-parameter fitting model.
- The calibration curve in Fig. 4d is plotted as (delta)SPR vs. log Albumin]. I would caution the authors against plotting signal vs. log concentration. A detailed discussion of why this is not best practice is provided in the recent Perspective article in Analytical Chemistry (P.L. Urban, Analytical Chemistry, 2020, 92, 10210-10212. DOI: 10.1021/acs.analchem.0c02096).
Answer: Dear reviewer, thanks for the comment. Indeed, the curve was not well defined with the data plotted. We added extra points to define the sigmoidal curve. We change the graphical 4d and the two graphics presented in the supplementary information (0%FBS y 10%FBS).
- There are a few typos that need to be corrected.
-Line 92: Testes?
-Line 105: “essay” should be “assay”
-Line 118: cm2. The “2” should be superscript
-Line 155 and elsewhere: N2. “2” should be subscript
-Line 266: “(2017, Biosens. Bioelectron.)” delete this parenthetical expression and cite Ref. [7].
-Line 191-192. Rewrite this sentence. As written it appears to indicate that the cell membrane contains fat droplets.
Answer: Dear reviewer, we have already fixed in the new version all the remarked typos.
Reviewer 3 Report
In the manuscript by Gerardo A. Lopez-Muñoz et al., authors developed a nanostructured plasmonic sensor based on commercial Blu-ray discs with integrated microfluidics for sensitive detection of albumin in cell culture supernatant from 2D fatty liver disease model. The manuscript does not have a significant impact and it is not particularly addressed of experiments, although seems well structured and easy to read. Therefore, only two things need to be adjusted:
- Figure 1 could better describe the image
- Section 3.1, from line 180: All the figures number mentioned in the text are referring to “Figure 1” which is obviously wrong, I believe they are referring to “Figure 2”
Author Response
In the manuscript by Gerardo A. Lopez-Muñoz et al., authors developed a nanostructured plasmonic sensor based on commercial Blu-ray discs with integrated microfluidics for sensitive detection of albumin in cell culture supernatant from 2D fatty liver disease model. The manuscript does not have a significant impact and it is not particularly addressed of experiments, although seems well structured and easy to read. Therefore, only two things need to be adjusted:
- Figure 1 could better describe the image
- Section 3.1, from line 180: All the figures number mentioned in the text are referring to “Figure 1” which is obviously wrong, I believe they are referring to “Figure 2”
Answer: Dear reviewer, we appreciate the comments; we have already fixed in the new version all the remarked typos.
Round 2
Reviewer 1 Report
Review of the article “Direct and Label-Free Monitoring of Albumin in 2D Fatty Liver Disease Model using Plasmonic Nanogratings” by G. A. Lopez-Muñoz, Ma. A. Ortega, A. Ferret-Miñana, F. De Chiara and J. Ramón-Azcón, Nanomaterials, Manuscript ID nanomaterials-1024759.
The authors replied satisfactorily to all of the questions and comments.
Minor comments:
- Line 46: please add a comma after “Therefore”.
- Lines 58, 74, 183, 207, 284, 301, 310, 311: in vitro should be in italics.
Author Response
- Line 46: please add a comma after “Therefore
- .”Lines 58, 74, 183, 207, 284, 301, 310, 311: in vitro should be in italics.
Answer: text modified.
Reviewer 2 Report
The authors have made a number of changes to address my earlier comments, however, one of their responses is not quite satisfactory. See below.
1. In Fig. 2b, the black arrows are still tiny. In fact at a quick glance they could be confused for cellular debris. Please make them larger. If you can’t make them larger, change the color. Black can be confused for debris. Also, what exactly are these arrows pointing at? On lines 188-190 the authors state,
“From day 1, all the experimental conditions displayed lipid accumulation using bright microscopy (black arrow, Figure 2b).”
What on these images indicates lipid accumulation? At this magnification there are no obvious differences. Because the differences must be quite subtle, they need to be fully explained. Tell the reader what features on the images indicate that lipid are being accumulated.
Author Response
Answer: Dear reviewer, thanks for your comments. We do apologize; it was a misunderstanding. We replaced the pictures and highlighted the details. We have detailed the explanation in the new version of the manuscript. We also added a bigger picture in the supplementary information.
"From day 1, all the experimental conditions displayed lipid droplets accumulation using bright microscopy (yellow arrows, Figure 2b). The NEFAs are internalized by the cells and processed as droplets of triglycerides (a zoomed image is shown in A1). "